# Going beyond Chat: Designing Connotative Meaningful Line Stickers to Promote Road Safety in Thailand through Participatory Design

Thawatphong Phithak , Pawanrat Surasangprasert and Sorachai Kamollimsakul *

Institute of Digital Arts and Science, Suranaree University of Technology, Nakhon Ratchasima 30000, Thailand; thawatphong@sut.ac.th (T.P.); pawanrat@g.sut.ac.th (P.S.)
* Correspondence: sorachai@sut.ac.th

**Abstract:** Road accidents are a leading cause of death in Thailand, with increasing fatalities. Despite road safety campaigns during holidays, consistent communication is lacking in daily life. This research aimed to create Line application stickers, a top chat platform for Thailand, using the participatory design (PD) approach. PD was implemented in two steps. Firstly, 100 participants outlined character types, moods, tones, and communication objectives. They recommended lively animal characters with diverse texts, such as greetings, work, travel, and emotions. Then, through a focus group, the tortoise was identified to represent cautious drivers who follow traffic rules, the rabbit to represent fast and risky drivers, and the zebra to represent vigilant and disciplined traffic police officers as characters for Line stickers. Subsequently, using the semiotics approach, 40 Line stickers were designed, and embedded with denotative and connotative road safety messages. Secondly, feedback from the focus group, integral to the PD process, led to refinements. After launching, a survey of 50 users showed "Benefits Received", "Text and Messages", and "Meaning" dimensions received "Very Satisfied/Strongly Agree" ratings. The "Character" dimension received a "Satisfied" rating. The results for "Benefits Received" can also be analyzed in the context of the Knowledge, Attitude, and Practice (KAP) theory, which revealed that K and A were at the highest level, while P was at a high level. This suggests that the Line stickers designed in this study effectively conveyed road safety messages to the receivers. This research constitutes the pioneering exploration within the realm of Line stickers concerning road safety, signifying the originality and unique contribution of our research to the existing body of knowledge in this domain. The PD process in this research can serve as a guideline for creating safety-promoting media in diverse fields.

**Keywords:** road safety; participatory design; online communication; social media

## 1. Introduction

The road accident situation and its management in Thailand are the major concerns that require resolution. Thailand still lacks a continuous coordination and operational role in addressing and monitoring these matters. There are also weak communications between the involved sectors to ensure accurate information dissemination to target groups, facilitating access to correct knowledge, and rigorously enforcing laws [1,2]. This situation is consistent with the cumulative statistics of the road accident fatalities in Thailand. According to the Thai Road Safety Collaboration, in the first quarter of the year 2023 [3], from January to March, the numbers have surpassed those of the year 2022 and are continuously increasing. In 2022, the cumulative numbers of the road accident-related deaths were 14,964 deaths.

Although Thailand has good law enforcement, there is still a lack of strict enforcement. Access to knowledge by the public is often conveyed through policies among sectors, campaigns during festivals or holidays, and primarily through organized activities. For instance, a study focused on enhancing road safety awareness among middle school

students through interactive learning emphasized creating safe road activity formats, fostering consciousness via experience sharing, and facilitating knowledge transfer and creativity. Such projects, while impactful, are tailored to a specific audience and timeframe, potentially limiting widespread knowledge dissemination. Hence, employing multiple methods and supplementary media are crucial to reach a broader audience [4].

Nowadays, social media has become a ubiquitous platform, with a global user base, and holds particular popularity as a mainstream medium in Thailand. Numerous studies have extensively examined the impact of social media on public issues, especially health communication and health promotion [5–7] and road safety [8,9]. A subset of research has delved into the influence of emoticons. Some previous research mentioned that incorporating emoticons into communication strategies proves to be a powerful tool for generating awareness and capturing the attention of target audiences [10]. Emoticons not only create and enhance awareness of messages but also contribute to improved communication within the online environment [11]. Furthermore, emoticons' presence in advertising communications has been demonstrated to elicit higher levels of positive effects among consumers, subsequently translating into increased purchase intentions [12,13]. This collective evidence highlights the influential role emotions play in fostering positive attitudes and behaviors. Wang [14] highlighted the distinctive nature of Line stickers: oversized emoticons that distinguish themselves by conveying intricate facial expressions and detailed body language. This unique feature enriches communication by providing heightened social and emotional cues, surpassing the capabilities of traditional emoticons. Despite the widespread use of Line, there has been limited research on Line stickers, with no prior studies specifically exploring their impact on road safety promotion. The prevalence of Line usage among Thai people suggests that Line stickers may have the potential to influence behavior across various social issues, particularly in the context of road safety.

In 2021, Thai people spent approximately 216 min per day on social media platforms. Specifically, Line application was the most widely used in Thailand, with an average of about 67 min per day, accounting for 31% of the total social media usage time. Additionally, the number of Line users in Thailand reached 50 million users, accounting for more than two out of three individuals in the country's population. The reasons for the popularity of Line, which was first introduced in Japan by a Korean company in 2011, were its ease of use for free instant messaging and user-friendliness and the Line sticker [15,16]. The use of Line stickers has been popular as they effectively enhance interactions among one-to-one and one-to-many communication [17–19]. Furthermore, Line Company Limited (Bangkok, Thailand) revealed that Thai users are among the world's highest purchasers of Line stickers, averaging 65 sticker sets per person, and this makes Thailand ranked in the top three countries globally with significant growth of sticker usage. Moreover, Thailand's creator market growth ranked in the first place, and the second position in the large creator base in Southeast Asia.

Sticker images within online chat platforms assume a significant role when considered within the framework of the Knowledge, Attitude, and Practice (KAP) theory [20]. In the context of knowledge acquisition, stickers offer a potent means of conveying information effectively through visually captivating and succinct representations. They serve as educational tools, imparting knowledge across diverse domains. This aids in augmenting users' awareness and comprehension of specific subjects. Furthermore, stickers extend their influence to shaping attitudes, as they transcend mere informational content. Employing emotive visuals and messages, stickers have the capacity to mold users' attitudes towards particular subjects. Importantly, stickers function as practical catalysts in the realm of behavioral modification. They act as reminders and motivators, prodding users towards the adoption of specific actions and habits. Consequently, stickers exhibit versatile capabilities that closely align with the core principles of the KAP theory, serving as valuable tools for disseminating knowledge, influencing attitudes, and prompting desired behavioral changes. These qualities underscore their significance in advocating for various causes within the digital communication milieu.

Incorporating the principles of the KAP theory, this study aimed to design Line stickers to promote road safety knowledge in Thailand. By using a participatory design procedure, the intention was not only to disseminate knowledge but also to influence attitudes and encourage safe practices. Additionally, the research evaluated user satisfaction and opinion with the designed stickers. The researcher aimed to seamlessly integrate road safety insights into daily life and online interactions through this initiative, filling this gap by investigating the potential knowledge, attitude, and behavior-changing effects of Line stickers, and shedding light on their role in the context of road safety awareness and promotion. Furthermore, the research methodology and communication insights derived from this study can also be applied as guidelines for designing and developing communication formats on various other social networks for campaigns and positive messaging.

## 2. Literature Review

### 2.1. Research Background

Thailand, renowned for its cultural richness and natural beauty, has also garnered attention for a less favorable reason: its high road traffic fatality rates. Historically, Thailand has frequently been cited by the World Health Organization (WHO) as one of the countries with the highest road traffic death rates per capita [21]. This alarming statistic is further underscored by the fact that a significant majority of these fatalities involve vulnerable road users, with motorcyclists being particularly affected [22].

Various studies have sought to understand the causes behind these accidents. A significant project spanning 2016–2020 by the Thailand Accident Research Center revealed that human factors were the primary contributors to motorcycle collisions. Failed perception emerged as the dominant cause, accounting for 49% of such accidents, followed by decision-making failure at 32%, and response failure at 13% [22]. In fatal accidents, failed perception was even more pronounced, contributing to 62% of the cases. These findings align with road accident research from various countries, indicating a global trend in the human factors leading to road mishaps [8,23–27]. In the urban context of Bangkok, a study highlighted several risk behaviors contributing to road accidents. The research pinpointed behaviors such as drunk driving, speeding, aggressive lane changing, and a general non-adherence to traffic rules as significant contributors to accidents [28].

The tourism sector, a cornerstone of Thailand's economy, is also impacted by this road safety crisis. Research focusing on foreign tourists renting motorcycles in Chiang Mai discovered that many people unfamiliar with local traffic laws and safety measures often rent without a valid license. This lack of awareness considerably increases the risk of severe accidents, emphasizing the need for multi-sectoral efforts to bolster sustainable tourism [29]. A broader perspective on road safety management in Thailand underscores that it is a collective endeavor. It involves not just the enforcement of traffic laws by governmental sectors but also policy formulation by local authorities and robust community engagement [30]. However, challenges remain. Altering the mindset and behavior of road users is a formidable challenge. Despite numerous awareness campaigns, many persist in risky behaviors, such as neglecting to wear helmets or driving under the influence. Moreover, while strict enforcement might be observed during high-risk periods, sustaining this level of vigilance year-round is essential for lasting change [31].

According to the previous research, one of the approaches that can contribute to promoting road safety awareness and cultivating a good sense of road safety is the spread of traffic-related information to enhance appropriate behaviors of the Thai population. The researcher recognized the significance of online communication through the Line application, which is the most popular online communication platform in Thailand, and decided to design and develop Line stickers to enhance knowledge about road safety in Thailand.

*2.2. Overview of the Popularity of Line Stickers in Thailand*

Line stickers have gained significant traction in Thailand, with both governmental and private sectors leveraging them to enhance information accessibility, advertisements, and public relations. There are primarily two categories of Line stickers: those available for sale, known as Line Creators Stickers; and Line Sponsored Stickers, which are available for free upon befriending the organization on the platform [31]. These stickers, often cartoonish in nature, depict various emotions, facial expressions, or convey short messages [32]. They offer multiple advantages, including establishing organizational identity, enhancing marketing memorability, facilitating direct communication with target audiences, and providing feedback through download statistics.

Research has shown that Line Sponsored Stickers, which are distributed for free, are primarily aimed at promoting sales. These stickers often introduce new cartoon characters that are playful, enjoyable, and gender-neutral. The perception of these stickers can effectively be linked to the brand they represent. For instance, the "MK Happy Duck" sticker set by MK Restaurants introduced a cartoon duck, symbolizing popular products like roasted duck [33]. Another notable example is the "Chao Lay" Line sticker set, designed to promote and preserve the Urak Lawoi language and culture. This sticker set garnered significant public interest, over 90.40%, due to its appealing design representing the Urak Lawoi community [34]. Furthermore, a study on designing character stickers for online social media emphasized the importance of diverse expressions, unique formats, distinctive personalities, appropriate sizes, modern language, and vibrant colors in the stickers [35].

The growing interest in Line stickers as a marketing tool has been highlighted in a study focusing on factors influencing Thai individuals in Bangkok to purchase Line stickers for communication [36]. Businesses have creatively utilized Line stickers for various purposes, including charitable causes. For instance, specialized Line sticker sets were designed to support disaster victims in the Philippines. The proceeds from these stickers, after deducting expenses, exceeded 14 million Thai Baht (THB), showcasing the potential of Line stickers as a contemporary communication tool. Another study delved into the semiotic significance of Line stickers [37]. It aimed to understand the relationships and intimacy levels between senders and recipients when communicating using these stickers. The findings revealed that Line stickers are not just tools for literal communication but also carry semiotic meanings. They can express emotions, convey closeness, and reflect the relationship dynamics between the sender and the recipient.

Based on the information provided, it is evident that previous research did not find any studies related to designing Line stickers for road safety campaigns. Therefore, this research has introduced participatory design guidelines for producing Line stickers. The insights gained from this research can also be applied to visual design for various other topics.

*2.3. Knowledge, Attitude, Practice (KAP)*

The Knowledge, Attitude, and Practice (KAP) model is a foundational framework predominantly utilized in the realms of public health, social sciences, and developmental projects to comprehend and evaluate the knowledge, attitudes, and practices of a specific community or group concerning a particular subject or intervention [38]. The "Knowledge" component delves into the cognitive understanding of individuals or communities about a topic, assessing what they have comprehended, heard, and are aware of. For instance, in campaigns centered around road safety, this facet would probe into the causes, consequences, and preventive measures related to road accidents. The "Attitude" dimension, on the other hand, explores the psychological inclinations of individuals or communities, shedding light on their beliefs, perceptions, and sentiments about a topic. In the context of road accidents, this would involve gauging the level of concern individuals harbor about being involved in an accident, their perceptions of its severity, and their disposition towards preventive measures like wearing seat belts or helmets. Lastly, the "Practice" segment pertains to the behavioral actions undertaken by individuals or communities in relation to the topic at hand. Using the road safety analogy, this would encompass behaviors like ad-

hering to speed limits, using pedestrian crossings, and avoiding distractions while driving. The merit of the KAP model lies in its ability to pinpoint knowledge gaps, misconceptions, and hindrances to behavioral change, thereby enabling researchers and policymakers to tailor interventions more adeptly to cater to the distinct needs of a community or group. However, it is imperative to acknowledge that human behavior is multifaceted, and while the KAP model offers a structured approach, it is often amalgamated with other models to ensure a holistic understanding [39].

In the domain of development communication, the KAP model serves as a foundational framework to understand the dynamics of knowledge dissemination and behavioral change. This model emphasizes the cognitive understanding (Knowledge), psychological inclinations (Attitude), and behavioral actions (Practice) of individuals or communities concerning specific topics [38]. For example, in the context of the COVID-19 pandemic, a study highlighted the impacts of knowledge, risk perception, emotion, and information on citizens' protective behaviors, emphasizing the role of official governmental media and the KAP model [40].

In the field of road safety, the KAP model has been extensively employed to understand and evaluate the behaviors and perceptions of drivers and road users. For instance, a study utilized the KAP model to assess the effects of mass communications on drinking and driving behaviors, highlighting the significance of passengers with a drunk driver or those charged with drunk driving [41]. A study conducted in Iran underscored the dominant role of drivers' attitudes in preventing road traffic crashes, emphasizing the issues of substance abuse and drunk driving [42]. Lastly, a roadside survey in Cambodia captured road users' KAPs regarding road safety, alcohol use, and drunk driving, providing valuable insights into the prevalent behaviors and attitudes in the region [43].

In addition to studying KAP to understand audience behavioral changes, research has also examined the impact of social networks in various contexts, including public health and online communication. For example, a review of 18 studies questioned Twitter's effectiveness in altering health behaviors. These studies highlighted Twitter's extensive reach and its limitations in driving meaningful public health communication, underlining the need for refined evaluation methods and consistent, long-term follow-ups in future health campaigns [5]. Moreover, organizations across different sectors are utilizing social media networks like Facebook and Twitter for stakeholder engagement, using them as cost-effective tools in their social marketing strategies. However, these strategies face constraints due to the interplay between technological artifacts and the embedded social structures within these platforms. The Puebla Sana program in Mexico is a case in point, demonstrating the complex interactions between technology and organizational strategies in the use of social media for health promotion [6]. Additionally, a comprehensive review of 28 studies focusing on social media's role in health promotion revealed varied methodologies. Platforms like Facebook and YouTube were used for interventions, while Twitter and Instagram served as tools for observing behaviors. These studies confirm the potential of social media in influencing behavior, yet they also point out challenges in measuring long-term behavioral changes. This suggests a need for campaigns that are tailored to specific stages of behavior change to enhance effectiveness [7].

### 2.4. Participatory Design (PD)

Participatory Design (PD) is a design methodology that actively involves all stakeholders, including end-users, in the design process to ensure that the product meets their needs and is usable. This approach is rooted in the belief that those who will be affected by a design should have a voice in making decisions about it [44]. Originating in Scandinavia in the 1970s, PD was initially a response to the labor movement's concerns about workers' rights and the impact of computerization on their jobs [45]. It was seen as a way to democratize the design process, giving a voice to those who would be most affected by the introduction of new technologies. Over time, the approach has been adapted and applied in various contexts, from software design to urban planning [46].

One of the primary benefits of PD is that it leads to products and solutions that are more aligned with users' actual needs. By involving users from the outset, designers can gain a deeper understanding of the context in which the product will be used, leading to more effective and user-friendly designs [47]. Moreover, the collaborative nature of PD fosters a sense of ownership among participants, increasing the likelihood of successful implementation and adoption [48].

PD has evolved as a significant approach in the realm of digital media and communication. With the integration of PD in digital media, such as social networking media, studies have adopted PD methods to examine methodologies for proposing new media designs [49]. The importance of visual learning, especially among the youth, has been underscored, suggesting that digital tools can enhance visual learning when designed through a participatory approach [50]. However, PD is not without its challenges. The process can be time-consuming and may require additional resources. There is also the potential for conflicts to arise among stakeholders with differing perspectives and priorities [51]. Despite these challenges, the benefits of creating more user-centered and contextually relevant designs often outweigh the drawbacks.

In conclusion, PD is a collaborative approach to design that prioritizes the involvement of all stakeholders, especially end-users, in the design process. By ensuring that those affected by a design have a say in its creation, PD aims to produce more effective, user-friendly, and contextually relevant solutions.

## 3. Research Objectives

### 3.1. Objective 1: Development of Line Stickers Utilizing a Participatory Design Approach

This research aims to develop visually engaging and informative stickers, specifically tailored for disseminating road safety knowledge in Thailand. The development process will adopt the PD methodology, engaging stakeholders in the design process to ensure the stickers are culturally relevant, informative, and appealing to the target audience. This objective aligns with the broader goal of enhancing public awareness and knowledge about road safety through innovative and accessible mediums.

### 3.2. Objective 2: Assessment of User Satisfaction/Opinion with the Designed Stickers

The second objective focuses on evaluating the effectiveness of the designed stickers from the perspective of user satisfaction/opinion. This assessment will be conducted through questionnaires, aiming to measure the level of satisfaction/opinion among users regarding the aesthetic appeal, informational content, and overall impact of the stickers in promoting road safety awareness.

## 4. Research Hypothesis

**H1.** *The developed stickers achieve a high level of user satisfaction/opinion in all dimensions.*

This hypothesis posits that the stickers designed for the road safety campaign in Thailand, utilizing the PD approach, will be well-received by users. It anticipates that users will rate their satisfaction and opinion with the stickers at a high level or above in all dimensions, indicating the effectiveness of the stickers in meeting their informational and aesthetic preferences and contributing positively to road safety awareness.

## 5. Research Methodology

This study utilizes the PD approach, involving users as stakeholders directly in the design process. Research activities were conducted to explore user needs, and upon data analysis, a series of Line stickers were designed based on the preferences of the target group. The researcher studied and gathered related information to design and develop Line stickers according to the following steps.

*5.1. Needs Analysis Employing Participatory Design*

This step involved factors influencing the effectiveness of PD consisting of (1) determining type of characters, characters' mood and tone, and communication goal; (2) determining specific characters, text, and phrases; and (3) determining denotative and connotative meaning for each sticker.

5.1.1. Determining Type of Characters, Characters' Mood and Tone, and Communication Goal

This study employed a PD approach to let the participants collaboratively define the needs for the Line stickers. The study was conducted according to the guidelines of the Declaration of Helsinki and approved by the Ethics Committee of Suranaree University of Technology on 8 November 2022 (COA No.93/2565). It screened volunteers for being over 15, holding a driver's license, and having interest in using social media for road safety promotion. Only those meeting all criteria were included in data collection. Data were collected by using an online survey with 100 participants who had valid driving licenses for any vehicle type in Thailand. All of them resided in Nakhon Ratchasima province, which in the year 2022 had the second-highest number of road traffic fatalities in Thailand, following Bangkok [52]. The participants were 54 males and 46 females, with ages ranging from 15 to 29 years. Occupationally, there were 62 government or state enterprise employees, 25 students, and 13 individuals from various other occupations. The survey consisted of three main topics: (1) types of characters, (2) communication goals, and (3) mood and tone (Figures 1–3).

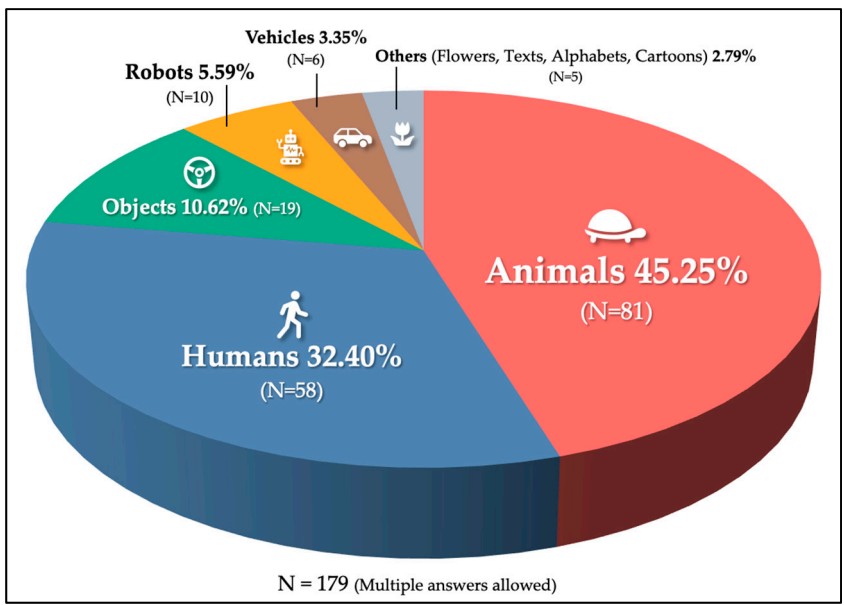

**Figure 1.** Types of characters survey results.

Figure 1 shows that animals were favored the most (45.3%), followed by humans (32.4%), and objects (10.6%). The results suggest that animal-like characters should be used for the stickers.

According to the results shown in Figure 2, the respondents favored a cute and lively theme the most (39%), followed by a realistic theme (24%), a minimal theme (18%), a photographic theme (13%), and other themes (6%). The conclusion drawn from these results is that the characters should be designed in a cute and lively theme.

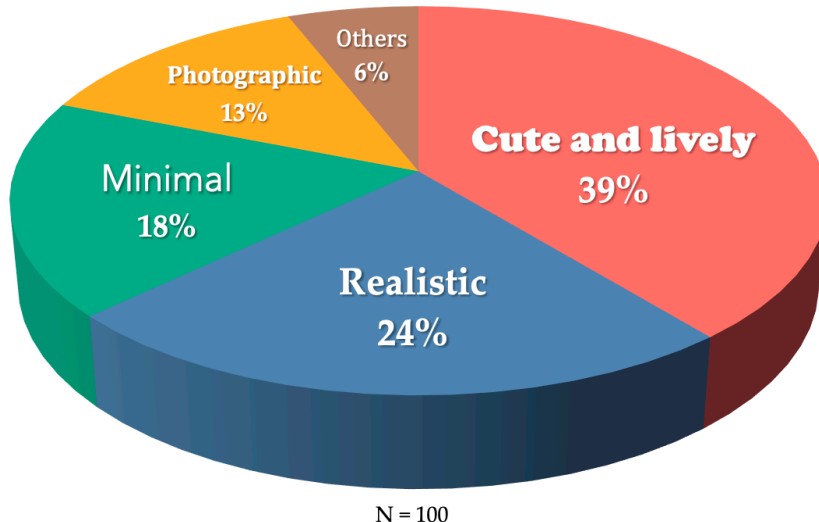

N = 100

**Figure 2.** Mood and tone of the characters survey results.

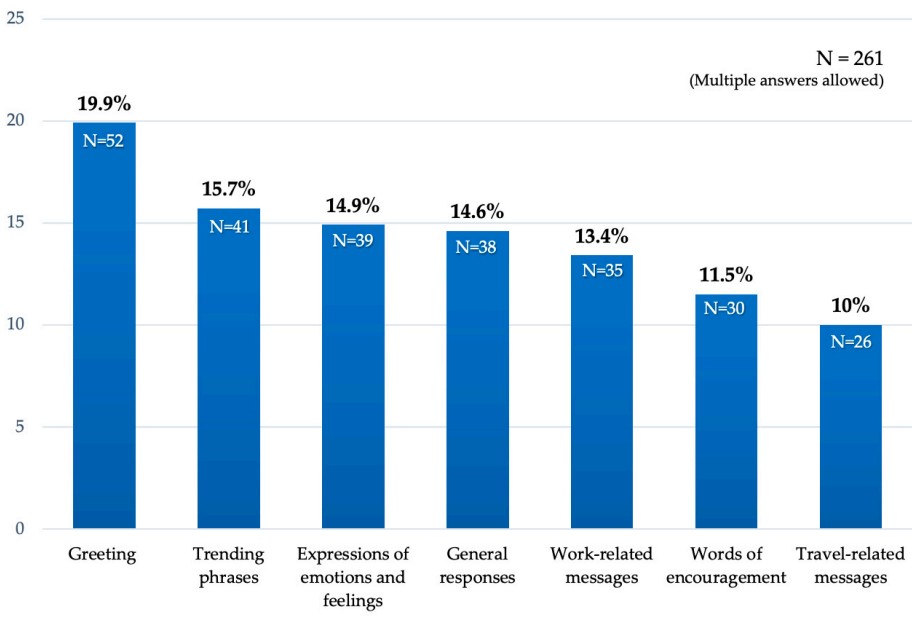

**Figure 3.** Communication goals survey results.

According to the findings in Figure 3, the respondents expressed a preference for various communication goals when utilizing stickers. The most popular choice was greetings at 19.9%, followed by messages related to trending phrases at 15.7%, and expressions of emotions and feelings at 14.9%. General responses were favored at 14.6%, with work-related messages coming in at 13.4%. Additionally, words of encouragement at 11.5%, and travel-related messages at 10% were also noted. The results were concluded and used for creating text and phrases for the subsequent sticker design.

5.1.2. Determining Specific Characters, Text, and Phrases

A focus group discussion was held with five participants, consisting of three males and two females. The discussion was divided into three sessions, each lasting for approximately 20 min. During the discussions, the researcher ensured that all participants had equal opportunities to express their opinions and that no one dominated the session. This was done to prevent any individual from unduly influencing or persuading others [53,54]. The discussion focused on two topics: (1) Which animal species could be representative of

promoting road safety in Thailand? (2) What text and phrases should be included in this sticker set? The outcomes of the focus group discussions were as follows:

1. To promote road safety in Thailand, three animals have been selected as representatives for the campaign's stickers. The tortoise represents cautious drivers who follow traffic rules, drive slowly but surely, and show empathy on the roads. The rabbit represents fast and risky drivers who tend to make hasty decisions and lack patience on the road. Both the tortoise and rabbit were drawn from Aesop's fable, "The Tortoise and the Hare". The zebra represents vigilant and disciplined traffic police officers who uphold regulations and respect rules. The zebra was chosen from the pedestrian crossing markings that consist of black and white stripes on roads, serving as a safe passage for pedestrians. These characters are presented in Figure 4.

2. The developed sticker set consisted of 40 messages, which is the maximum number feasible for a single image sticker set. There were seven categories of messages as presented in Table 1.

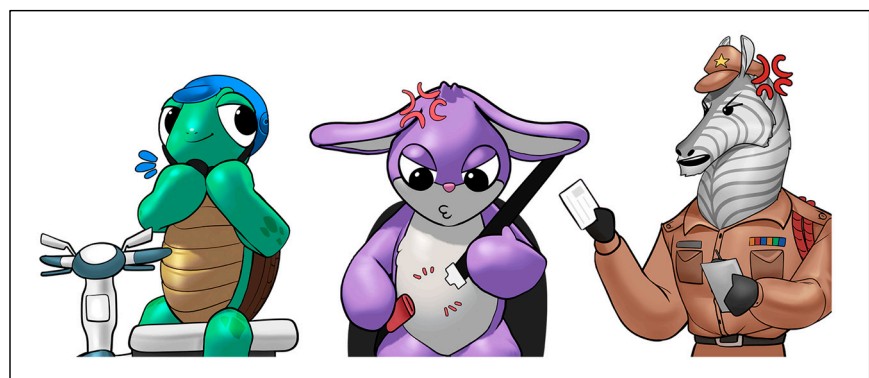

**Figure 4.** The designed tortoise, rabbit, and zebra.

**Table 1.** Text and messages in the developed Line stickers.

| Categories | Communication Goals | N (Messages) | Texts/Messages |
|:---:|:---:|:---:|:---:|
| 1 | Greeting | 4 | (1) Hello (2) Bye (3) Good night (4) Let's hangout |
| 2 | Trending phrases | 8 | (1) Big bros! (2) Calm down! (3) Fighting! (4) Don't play with fire! (5) How dare you! (6) So drama! (7) Get high! (8) Smell fishy! |
| 3 | Expressions of emotions and feelings | 9 | (1) Hahaha (2) Hehe (3) Just kidding (4) Give up! (5) Offensive! (6) Be careful! (7) I wanna cry (8) In hurry? (9) Exhausted! |
| 4 | General responses | 5 | (1) Okay (2) Thank you (3) Roger that (4) Ready! (5) Happy birthday |
| 5 | Work-related messages | 4 | (1) Focus! (2) Keep calm! (3) Safety first! (4) Be cautious! |
| 6 | Words of encouragement | 4 | (1) Take care! (2) Sorry (3) Love you (4) Congratulations |
| 7 | Travel-related messages | 6 | (1) Safe trip (2) Don't drive drunk (3) Stop (4) Be generous (5) Keep distance (6) Got lost! |

### 5.1.3. Determining Denotative and Connotative Meaning for Each Sticker

This step was conducted to develop the results as the inputs into Line stickers that can be used for everyday conversations and also integrated with the knowledge about road safety. The researcher introduced the concept of semiotics [55,56] in the design to the same group of participants, aiming to make an accurate understanding of the meaning behind the designed sticker. The key content consisted of two types of meanings. (1) Denotative meaning, which is the direct and explicit meaning that people generally understand, aligning with the visible text or images. It is in the descriptive level that is accessible

and comprehensible to most recipients. (2) Connotative meaning, which goes beyond the literal interpretation. It varies from person to person and emerges when the denotative meaning is associated with the implied meaning. For example, an image of a tortoise character driving a car to let an ambulance pass (denotative meaning) conveys the idea of empathy and mutual consideration (connotative meaning). Another example is an image of a drunk-driving rabbit with a crossed-out symbol (denotative meaning), that implies the legal offense of driving under the influence (connotative meaning), and so on. Through determining the roles of participation, the results in the stickers embedded with meanings related to road safety are illustrated in Table 2.

**Table 2.** Examples of Line stickers for promoting road safety.

| Texts/ Messages | Images | Denotative Meaning | Connotative Meaning | Texts/ Messages | Images | Denotative Meaning | Connotative Meaning |
|---|---|---|---|---|---|---|---|
| 1. Thank you! | | A tortoise is walking across the zebra crossing. Cars are stopped to let it pass. | Expressing gratitude when receiving assistance from others. | 7. Ready! | | The tortoise is inspecting a vehicle that is in a usable condition. | Being well-prepared before travelling. |
| 2. Keep calm! | | The tortoise is comfortably driving the car without rushing. | Reminding yourself to perform actions gradually and steadily. | 8. Safety first! | | The tortoise is wearing a helmet before driving. | Preventing danger before engaging in any activity. |
| 3. How dare you! | | A rabbit is parking the car in a white and red parking area. | Maintaining discipline and refraining from inappropriate actions. | 9. Focus! | | The tortoise, inside the car, is saying, "focus," to concentrate before pressing the car's start button. | Being focused in the present, controlling emotions and feelings steadfastly. |
| 4. Don't drive drunk | | The rabbit is driving drunk. There is a sign of intoxication. | Driving a vehicle drunk is illegal. | 10. Be careful! | | The rabbit suddenly is hitting brake, responsible for blocking the railroad crossing. | Taking care without negligence to prevent harm. |
| 5. Don't play with fire! | | The rabbit is driving through a red light while the straight-going cars are exiting. | Avoiding risks with known potential dangers. | 11. Be generous | | The tortoise is driving the car to let the ambulance pass. | Being generous and understanding, allowing mutual compassion. |
| 6. Calm down! | | The rabbit is driving at high speed, overtaking the tortoise. | Reducing excessive behaviors. | 12. Keep distance | | Two cars are driving, maintaining distance between them. | Maintaining a gap between personal relationships, not letting them become too close to each other. |

## 5.2. Line Stickers' Development Employing Participatory Design

The Line stickers were carried out using Clip Studio Paint Pro software version 2.0 (CELSYS, Inc., Tokyo, Japan) for drawing the draft, coloring, and arranging texts and messages in Thai. After that, all images were provided to the same group of participants involved in the group discussion to give feedback, opinions, and additional suggestions. The stickers were edited following the feedback and suggestions received before launching this set of stickers in the Line Store. Figure 5 illustrates the conceptual sketches of the three characters, developed through the participatory design (PD) process. The sticker set was named "Roady and Safety" with Roady being the name of the rabbit character, and Safety being the name of the tortoise character. The stickers were 40 static image stickers with dimensions of 370 pixels in width and 320 pixels in height. These stickers can be downloaded at https://store.line.me/stickershop/product/23957699 (accessed on 15 November 2023) (Figure 6).

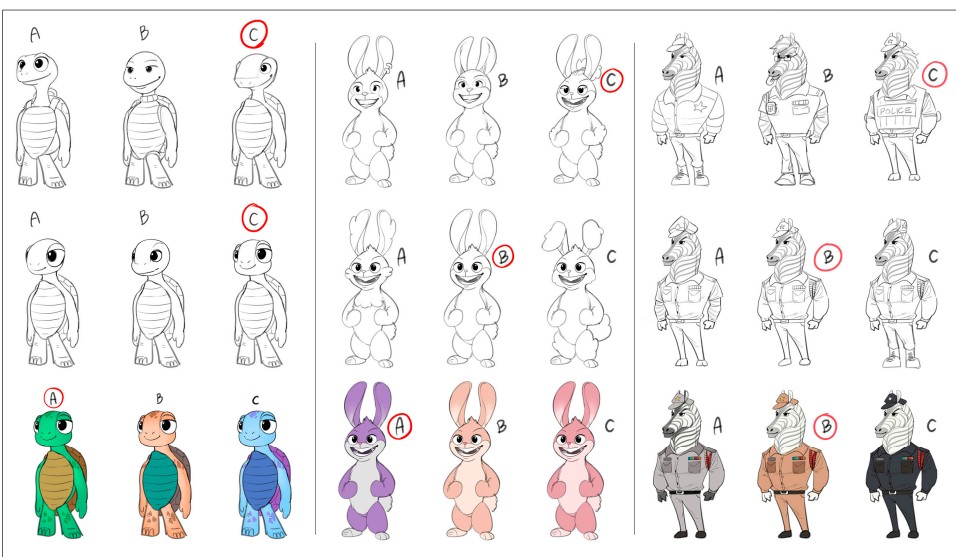

**Figure 5.** The conceptual sketches of the three characters.

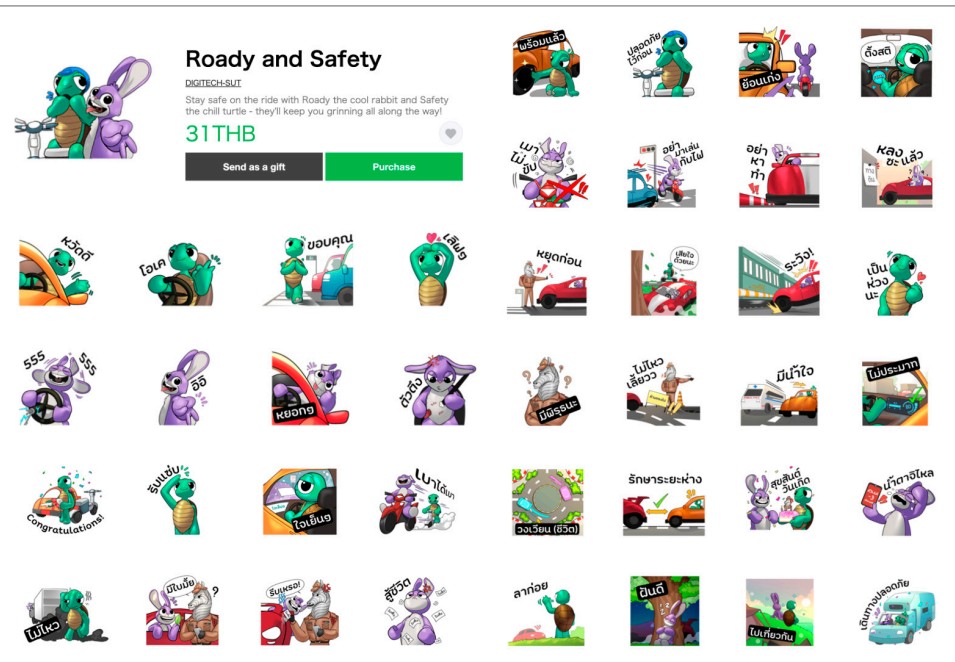

**Figure 6.** The designed stickers download page on the Line Store.

### 5.3. Evaluation Step

The evaluation questionnaire, which relates to the assessment of user satisfaction and opinions, was conducted using a 5-point Likert scale, following the guidelines established by Likert [57], as shown in Table 3.

**Table 3.** User satisfaction and opinion evaluation scale.

| Scale | Scale Interval | Level | Scale Description | |
| --- | --- | --- | --- | --- |
| | | | User Satisfaction Evaluation * | User Opinion Evaluation ** |
| 5 | 4.50–5.00 | Highest | Very Satisfied | Strongly Agree |
| 4 | 3.50–4.49 | High | Satisfied | Agree |
| 3 | 2.50–3.49 | Moderate | Acceptable | Neutral/Uncertain |
| 2 | 1.50–2.49 | Low | Dissatisfied | Disagree |
| 1 | 1.00–1.49 | Lowest | Very Dissatisfied | Strongly Disagree |

* Evaluation of user satisfaction is used to assess the character dimension (questions 1.1–1.5), text and message dimension (questions 2.1–2.6), meaning dimension (questions 3.1–3.7), and benefits received dimension (questions 4.1–4.2). ** The evaluation of user opinions is used to assess the benefits received dimension consistent with the KAP theory (questions 4.3–4.5).

The satisfaction evaluation questionnaire was designed to measure four dimensions: (1) character, (2) text and message, (3) meaning, and (4) benefits received. To ensure the questionnaire's validity, three experts used the Index of Item Objective Congruence (IOC) method [58] to validate it. The IOC index for the questionnaire was 0.934, which indicates that the questionnaire items are aligned with the research objectives. A reliability analysis using Cronbach's alpha was conducted to assess the consistency of responses to the scale items. The computed alpha coefficient of 0.956 indicates a high level of internal consistency among the questionnaire questions.

The designed Line stickers were launched to the target participants, which consisted of 50 individuals holding driver's licenses in Thailand. Among them, 46% were males and 54% were females. Furthermore, 26% were under 20 years old, 56% were between 20 and 29 years old, 12% were between 30 and 39 years old, and 6% were 40 years old and above. The participants used the Line stickers for 1 week and responded to the satisfaction and opinion evaluation questionnaire.

## 6. Research Results

The results of the satisfaction and opinion questionnaires are presented in Tables 4 and 5.

Table 4 reveals that a two-sample *t*-test was conducted to compare the satisfaction/opinion of the Line stickers between the male and female groups, revealing no significant difference in satisfaction/opinion ratings for the Line stickers ($p > 0.05$). Moreover, a one-way ANOVA was performed to compare the effect of four different age groups on satisfaction/opinion with the Line stickers. A one-way ANOVA revealed no statistically significant difference in satisfaction/opinion ratings of the Line stickers for at least two groups ($p > 0.05$). This implied that the Roady and Safety Line stickers are suitable for both males and females in any age group.

**Table 4.** T-test difference of gender and satisfaction/opinion and ANOVA difference of age group and satisfaction/opinion.

| Dimension | Criteria | Male $\bar{x}$ | Male S.D. | Female $\bar{x}$ | Female S.D. | $t(48)$ | $p$ | 15–20 $\bar{x}$ | 15–20 S.D. | 20–29 $\bar{x}$ | 20–29 S.D. | 30–39 $\bar{x}$ | 30–39 S.D. | 40 and Above $\bar{x}$ | 40 and Above S.D. | $F$ (3.49) |
|---|---|---|---|---|---|---|---|---|---|---|---|---|---|---|---|---|
| 1 | Character Dimension | | | | | | | | | | | | | | | |
| 1.1 | Attractiveness of the characters | 4.30 | 0.76 | 4.37 | 4.33 | −0.335 | 0.739 | 4.23 | 0.83 | 4.39 | 0.69 | 4.17 | 0.41 | 4.67 | 0.58 | 0.50 |
| 1.2 | Proportions and size of the characters | 4.39 | 0.72 | 4.56 | 0.74 | −0.852 | 0.398 | 4.54 | 0.78 | 4.43 | 0.63 | 4.50 | 0.84 | 4.67 | 0.58 | 0.16 |
| 1.3 | Uniqueness and distinctiveness of the characters | 4.22 | 0.74 | 4.59 | 0.69 | −1.853 | 0.070 | 4.23 | 0.83 | 4.46 | 0.69 | 4.50 | 0.84 | 4.67 | 0.58 | 0.48 |
| 1.4 | Color tones used | 4.08 | 0.72 | 4.30 | 0.61 | −1.103 | 0.275 | 3.92 | 0.64 | 4.21 | 0.63 | 4.50 | 0.84 | 4.67 | 0.58 | 1.70 |
| 1.5 | Overall likability of the characters | 4.22 | 0.67 | 4.37 | 0.63 | −0.831 | 0.410 | 4.15 | 0.69 | 4.28 | 0.66 | 4.50 | 0.55 | 4.67 | 0.58 | 0.73 |
| | Average | 4.27 | 0.54 | 4.40 | 0.44 | −0.914 | 0.365 | 4.25 | 0.52 | 4.34 | 0.46 | 4.34 | 0.58 | 4.70 | 0.44 | 0.67 |
| 2 | Text and Message Dimension | | | | | | | | | | | | | | | |
| 2.1 | Clarity of the text | 4.48 | 0.51 | 4.67 | 0.51 | −1.343 | 0.186 | 4.62 | 0.51 | 4.61 | 0.50 | 4.33 | 0.52 | 4.67 | 0.58 | 0.55 |
| 2.2 | Ease of use (such as reduced typing time) | 4.30 | 0.72 | 4.37 | 0.48 | −0.309 | 0.758 | 4.00 | 0.58 | 4.43 | 0.79 | 4.50 | 0.84 | 4.67 | 0.58 | 1.35 |
| 2.3 | Font type | 4.39 | 0.66 | 4.48 | 0.64 | −0.490 | 0.627 | 4.23 | 0.73 | 4.48 | 0.64 | 4.50 | 0.55 | 5.00 | 0.00 | 1.26 |
| 2.4 | Font size | 4.30 | 0.70 | 4.59 | 0.50 | −1.688 | 0.098 | 4.38 | 0.51 | 4.54 | 0.64 | 4.17 | 0.75 | 4.67 | 0.58 | 0.77 |
| 2.5 | Text arrangement and positioning | 4.60 | 0.50 | 4.56 | 0.64 | −0.323 | 0.748 | 4.62 | 0.51 | 4.57 | 0.63 | 4.50 | 0.55 | 4.67 | 0.58 | 0.08 |
| 2.6 | Text and background colors | 4.60 | 0.50 | 4.56 | 0.64 | 0.323 | 0.748 | 4.38 | 0.65 | 4.64 | 0.56 | 4.50 | 0.55 | 5.00 | 0.00 | 1.20 |
| 2.7 | Trending phrases | 4.52 | 0.67 | 4.56 | 0.80 | −0.161 | 0.873 | 4.46 | 0.78 | 4.54 | 0.79 | 4.67 | 0.52 | 4.67 | 0.58 | 0.13 |
| | Average | 4.46 | 0.47 | 4.54 | 0.53 | −0.584 | 0.562 | 4.38 | 0.41 | 4.65 | 0.57 | 4.45 | 0.41 | 4.77 | 0.40 | 0.61 |
| 3 | Meaning Dimension | | | | | | | | | | | | | | | |
| 3.1 | Facial expressions and postures of the "tortoise" character | 4.35 | 0.78 | 4.48 | 0.51 | −0.730 | 0.469 | 4.46 | 0.66 | 4.36 | 0.68 | 4.50 | 0.55 | 4.67 | 0.58 | 0.27 |
| 3.2 | Facial expressions and postures of the "rabbit" character | 4.48 | 0.73 | 4.59 | 0.57 | −0.620 | 0.538 | 4.38 | 0.77 | 4.61 | 0.63 | 4.50 | 0.55 | 4.67 | 0.58 | 0.38 |
| 3.3 | Facial expressions and postures of the "zebra" character | 4.52 | 0.73 | 4.37 | 0.63 | 0.787 | 0.435 | 4.46 | 0.66 | 4.39 | 4.74 | 4.33 | 0.52 | 5.00 | 0.00 | 0.78 |

**Table 4.** *Cont.*

| Dimension | Criteria | Gender | | | | | | Age Group | | | | | | | | *F* |
| | | Male | | Female | | *t*(48) | *p* | 15–20 | | 20–29 | | 30–39 | | 40 and Above | | (3.49) |
| | | $\bar{x}$ | S.D. | $\bar{x}$ | S.D. | | | $\bar{x}$ | S.D. | $\bar{x}$ | S.D. | $\bar{x}$ | S.D. | $\bar{x}$ | S.D. | |
| 3.4 | Interpretation of other elements in the picture such as background scenes, vehicles, atmosphere | 4.65 | 0.57 | 4.48 | 0.58 | 0.433 | 0.302 | 4.84 | 0.38 | 4.39 | 0.63 | 4.50 | 0.55 | 5.00 | 0.00 | 2.71 |
| 3.5 | Conveying feelings to others through the use of these Line stickers | 4.39 | 0.72 | 4.44 | 0.70 | −0.264 | 0.793 | 4.31 | 0.63 | 4.43 | 0.79 | 4.50 | 0.55 | 4.67 | 0.58 | 0.25 |
| 3.6 | Interest level in the story depicted in these Line stickers | 4.39 | 0.78 | 4.51 | 0.70 | −0.607 | 0.547 | 4.54 | 0.52 | 4.39 | 0.83 | 4.33 | 0.82 | 5.00 | 0.00 | 0.72 |
| 3.7 | Overall integration of knowledge in promoting road safety within these Line stickers | 4.65 | 0.65 | 4.67 | 0.48 | −0.091 | 0.928 | 4.85 | 0.38 | 4.61 | 0.63 | 4.50 | 0.55 | 4.67 | 0.58 | 0.72 |
| | Average | 4.49 | 0.59 | 4.51 | 0.44 | −0.166 | 0.869 | 4.55 | 0.41 | 4.46 | 0.58 | 4.43 | 0.42 | 4.80 | 0.35 | 0.48 |
| 4 | Benefits Received Dimension | | | | | | | | | | | | | | | |
| 4.1 | Suitability of Line stickers for daily communication | 4.39 | 0.72 | 4.33 | 0.62 | 0.305 | 0.761 | 4.15 | 0.55 | 4.43 | 0.69 | 4.33 | 0.81 | 4.67 | 0.58 | 0.72 |
| 4.2 | Appropriate quantity of 40 stickers for use | 4.61 | 0.66 | 4.67 | 0.62 | −3.21 | 0.750 | 4.53 | 0.66 | 4.64 | 0.62 | 4.67 | 0.82 | 5.00 | 0.00 | 0.43 |
| 4.3 | Promoting road safety understanding in Thailand (Knowledge) | 4.65 | 0.49 | 4.70 | 0.47 | −0.382 | 0.704 | 4.53 | 0.52 | 4.68 | 0.48 | 4.83 | 0.41 | 5.00 | 0.00 | 1.07 |
| 4.4 | Raising awareness of safe land transportation (Attitude) | 4.52 | 0.67 | 4.62 | 0.56 | −0.620 | 0.538 | 4.46 | 0.52 | 4.64 | 0.62 | 4.33 | 0.82 | 5.00 | 0.00 | 1.07 |
| 4.5 | Reinforcing traffic discipline for safety (Practice) | 4.26 | 0.69 | 4.59 | 0.57 | −1.860 | 0.069 | 4.38 | 0.77 | 4.50 | 0.58 | 4.33 | 0.82 | 4.33 | 0.58 | 0.19 |
| | Average | 4.49 | 0.46 | 4.59 | 0.35 | −0.862 | 0.393 | 4.42 | 0.34 | 4.58 | 0.41 | 4.50 | 0.67 | 4.80 | 0.41 | 0.94 |

**Table 5.** The results of satisfaction/opinion on the use of Roady and Safety Line stickers.

| Dimension | Criteria | x̄ | S.D. | Description |
|---|---|---|---|---|
| 1 | Character Dimension | | | |
| 1.1 | Attractiveness of the characters | 4.33 | 0.61 | Satisfied |
| 1.2 | Proportions and size of the characters | 4.47 | 0.60 | Satisfied |
| 1.3 | Uniqueness and distinctiveness of the characters | 4.43 | 0.63 | Satisfied |
| 1.4 | Color tones used | 4.17 | 0.62 | Satisfied |
| 1.5 | Overall likability of the characters | 4.29 | 0.58 | Satisfied |
| | | 4.34 | 0.61 | Satisfied |
| 2 | Text and Message Dimension | | | |
| 2.1 | Clarity of the text | 4.58 | 0.50 | Very Satisfied |
| 2.2 | Ease of use (such as reduced typing time) | 4.34 | 0.75 | Satisfied |
| 2.3 | Font type | 4.44 | 0.64 | Satisfied |
| 2.4 | Font size | 4.46 | 0.61 | Satisfied |
| 2.5 | Text arrangement and positioning | 4.58 | 0.57 | Very Satisfied |
| 2.6 | Text and background colors | 4.58 | 0.57 | Very Satisfied |
| 2.7 | Trending phrases | 4.54 | 0.73 | Very Satisfied |
| | Average | 4.50 | 0.63 | Very Satisfied |
| 3 | Meaning Dimension | | | |
| 3.1 | Facial expressions and postures of the "tortoise" character | 4.42 | 0.64 | Satisfied |
| 3.2 | Facial expressions and postures of the "rabbit" character | 4.54 | 0.65 | Very Satisfied |
| 3.3 | Facial expressions and postures of the "zebra" character | 4.44 | 0.67 | Satisfied |
| 3.4 | Interpretation of other elements in the picture such as background scenes, vehicles, atmosphere | 4.56 | 0.58 | Very Satisfied |
| 3.5 | Conveying feelings to others through the use of these Line stickers | 4.42 | 0.70 | Satisfied |
| 3.6 | Interest level in the story depicted in these Line stickers | 4.46 | 0.73 | Satisfied |
| 3.7 | Overall integration of knowledge in promoting road safety within these Line stickers | 4.66 | 0.56 | Very Satisfied |
| | Average | 4.50 | 0.65 | Very Satisfied |
| 4 | Benefits Received Dimension * | | | |
| 4.1 | Suitability of Line stickers for daily communication | 4.36 | 0.66 | Satisfied |
| 4.2 | Appropriate quantity of 40 stickers for use | 4.64 | 0.63 | Very Satisfied |
| 4.3 | Promoting road safety understanding in Thailand (Knowledge) | 4.68 | 0.47 | Strongly Agree |
| 4.4 | Raising awareness of safe land transportation (Attitude) | 4.58 | 0.61 | Strongly Agree |
| 4.5 | Reinforcing traffic discipline for safety (Practice) | 4.44 | 0.64 | Agree |
| | Average | 4.54 | 0.60 | Very Satisfied, Strongly Agree |

* Questions 4.1–4.2. were used to assess the satisfaction on the benefits received dimension. Questions 4.3–4.5 were used to assess the opinion on the benefits received dimension consistent with the KAP theory.

In Table 5, the results of the satisfaction evaluation for the usage of the "Roady and Safety" Line sticker set, which integrated knowledge about road safety in Thailand and was

designed with a PD approach, showed that users expressed varying levels of satisfaction across different dimensions. The "Benefits Received" dimension ($\bar{x}$ = 4.54, S.D. = 0.60), the "Text and Messages" dimension ($\bar{x}$ = 4.50, S.D. = 0.63), and the "Meaning" dimension ($\bar{x}$ = 4.50, S.D. = 0.65) received the Very Satisfied/Strongly Agree levels. The "Character" dimension ($\bar{x}$ = 4.34, S.D. = 0.61) received a Satisfied level.

## 7. Discussion and Conclusions

This research utilized a participatory design (PD) for Line stickers. The two key steps were: Step 1, a needs analysis; and Step 2, Line sticker development.

In Step 1, the needs analysis employing PD, the participants determined character type, mood, tone, and communication goals via online questionnaires. The results show that the participants favored cute animals with greetings, emotions, work, encouragement, and travel messages. In addition, specific characters and phrases were identified through focus groups with five participants. Animals like tortoises, rabbits, and zebras represented road safety themes, with a suggestion of 40 stickers per set. Moreover, the participants were introduced to semiology theory, and then they were asked to determine denotative and connotative meanings in each sticker.

In Step 2, Line sticker development employing PD, all the data gathered from the previous step were used to develop a set of static image stickers called "Roady and Safety". The participants from the previous focus group provided feedback and suggestions. The designer edited each sticker until all five participants approved. The outcome was 40 Line stickers for daily communication, emphasizing road safety.

After that, the target group was asked to use the stickers, followed by responding to the satisfaction evaluation questionnaire. The "Benefits Received" dimension, the "Text and Messages" dimension, and the "Meaning" dimension received the Very Satisfied/Strongly Agree levels. The "Character" dimension received a Satisfied level. Consequently, the research hypothesis H1 is accepted. These findings indicate that the stickers, designed for the Thai road safety campaign, effectively met user expectations in terms of information and design, thereby achieving a high level of user satisfaction.

The results also strongly confirmed the benefit of utilizing the Participatory Design (PD) approach [47–49]. The stickers not only met but surpassed user satisfaction, contributing significantly to both informational content and aesthetic preferences, thereby positively impacting road safety awareness.

In addition, the questions presented in the "Benefits Received" assessment are aligned with the KAP theory, which demonstrates the relationship between Knowledge, Attitude, and Practice. Users expressed their highest agreement (Strongly Agree) regarding questions related to Knowledge and Attitude, while the question linked to Practice received a high level of agreement (Agree). This was in line with the theory that changing Practice is harder than Attitude and changing Attitude is harder than Knowledge. Moreover, this alignment with the KAP theory suggests a positive influence of the stickers on users' awareness, attitudes, and potential behavioral practices related to road safety. Figure 7 shows the average score of user opinion in the KAP dimensions for Line stickers.

According to the KAP theory, the actions of the recipients regarding their surroundings are influenced by various factors within society, particularly from the social system and media systems. This aligns with the communication format through the Line application, where, when recipients receive stickers embedded with implied meanings related to road safety from acquaintances, it leads to the formation of positive attitudes and an increased willingness to change driving behaviors to be more cautious. Figure 8 illustrates the components of KAP that occur between the sender and receiver through communication using Line stickers. It is evident that the research guidelines can be applied to design and develop online communication media. The application of the insights derived from this research, categorized according to the components of communication, is as follows:

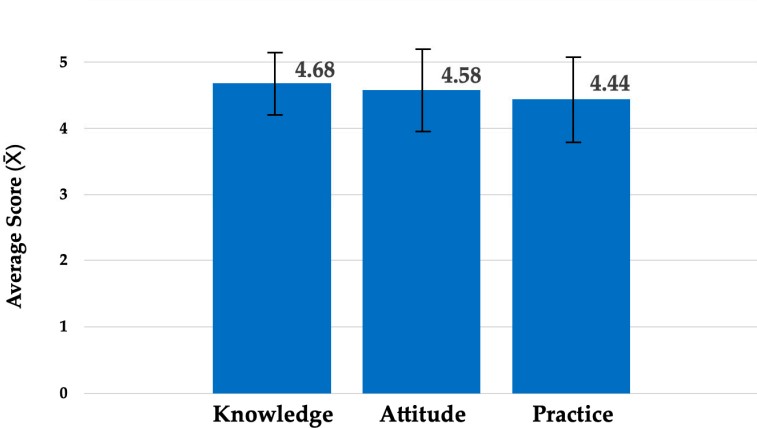

**Figure 7.** The average score of user opinion in KAP dimensions for Line stickers.

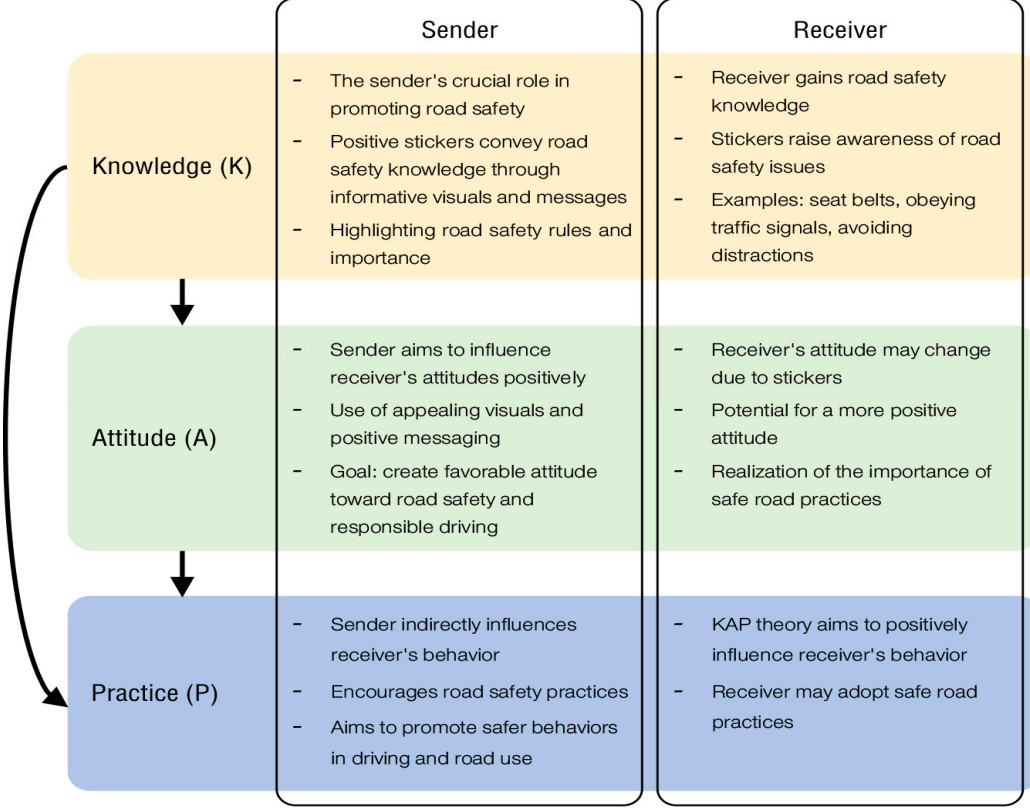

**Figure 8.** The components of KAP that occur between the sender and receiver through communication using Line stickers.

### 7.1. Sender or Source

The sender is the person or entity initiating the communication. In this context, the sender can be an individual, a road safety organization, a government agency, or any entity interested in promoting road safety. The sender is responsible for creating or selecting the stickers related to road safety and using them to convey a message. Their role is to disseminate information and influence the receiver's knowledge, attitudes, and practices related to road safety through the stickers.

To ensure the widespread use and public benefit from the developed Line stickers, the research team has planned two dissemination strategies: (1) on-site distribution, which involves handing out stickers during traffic-related training sessions at the university; and (2) online distribution through the university's social networks. Interested individuals can

download the stickers from the Line Store in Thailand by searching for 'Roady and Safety' at a cost of THB 31 per set. As of now, there have been approximately 300 downloads (data from 1 October to 15 November 2023).

### 7.2. Message

The message refers to the content being communicated. In this case, it consists of the stickers related to road safety. The stickers can convey various aspects of road safety, such as wearing seat belts, obeying traffic rules, avoiding distractions while driving, or showing empathy and support for safe driving practices. The effectiveness of the message lies in its ability to capture the receiver's attention, convey a clear road safety message, and elicit a positive response.

### 7.3. Channel

The channel represents the medium or platform through which the message is transmitted from the sender to the receiver. In this scenario, Line application serves as a convenient and accessible channel for delivering road safety messages in the form of stickers.

### 7.4. Receiver

The receiver is the individual or group intended to receive and interpret the message. In this context, it can be any user of the Line application who receives the road safety stickers. The receiver's role is to process and interpret the stickers, which can influence their knowledge (what they learn about road safety), attitudes (how they feel about road safety), and practices (how they behave on the road). The ultimate goal is to encourage the receiver to adopt safer road practices and promote a positive road safety culture.

The findings of our research affirm the noteworthy influence of Line stickers on communication, aligning seamlessly with prior studies that underscore the role of emoticons in enhancing satisfaction, enjoyment, and fostering increased information sharing. These outcomes are consistent with the existing literature, as evidenced by previous research [10–13,17]. Participants in our study expressed their agreement on the positive impact of Line stickers on knowledge, attitude, and practice. This can be attributed to the unique quality of Line stickers, which operate as oversized emoticons processed in emotional centers, surpassing the engagement levels achieved by words primarily processed through logical channels. Furthermore, the proficiency of Line stickers in conveying intricate facial expressions and detailed body language contributes to elevated social and emotional cues, thereby enriching communication experiences [14].

This study has interesting findings but also has some limitations. The stickers resulting from this research have been specifically designed for users in Thailand, featuring Thai language text within the images. In future research, the research team is interested in exploring textless image design to make it accessible without language limitations. Furthermore, they aim to apply the user-participatory design guidelines derived from this study to create various other types of media under compelling topics that can benefit individuals within society.

**Author Contributions:** Conceptualization, T.P. and P.S.; Methodology, T.P. and S.K.; Software, T.P.; Validation, S.K.; Formal Analysis, T.P. and S.K.; Investigation, P.S.; Resources, T.P. and S.K.; Data Curation, T.P. and S.K.; Writing—Original Draft, T.P.; Writing—Review and Editing, S.K.; Visualization, P.S.; Supervision, T.P. and S.K.; Project Administration, T.P.; Funding Acquisition, T.P. and S.K. All authors have read and agreed to the published version of the manuscript.

**Funding:** This research was funded by (i) Suranaree University of Technology (SUT), (ii) Thailand Science Research and Innovation (TSRI), and (iii) National Science, Research, and Innovation Fund (NRSF) [Project code: 4284945].

**Institutional Review Board Statement:** The study was conducted according to the guidelines of the Declaration of Helsinki and approved by the Ethics Committee of Suranaree University of Technology on 8 November 2022 (COA No.93/2565).

**Informed Consent Statement:** Not applicable.

**Data Availability Statement:** Data are contained within the article.

**Conflicts of Interest:** The authors declare no conflict of interest.

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
