# Peer review of "Going beyond Chat: Designing Connotative Meaningful Line Stickers to Promote Road Safety in Thailand through Participatory Design"

_safety, 2023_

Round 1

Reviewer 1 Report

Comments and Suggestions for Authors

This paper uses online stickers to promote road safety, it seems interesting and useful. However, the overall innovation of the article is insufficient. And the connection between stickers and safety is inconspicuous.

1.Introduction and Literature Review

In Introduction, the background has been explained. But more details about how these designs affect safety have not been given. Please add some references to illustrate the necessary of this reasearch.

Although the authors evaluated user satisfaction through a questionnaire survey. However, the questionnaire has the following issues:

4. Research Results 

1. The author found that the questions related to Knowledge and Attention were rated at the highest level (Very Satisfied) while the question related to Practice was rated at a high level (Satisfied) This was in line with the theory that changing practice is hard than attribute and changing attribute is hard than knowledge This is a conclusion of the article. To increase credibility, I suggest the author provide a bar chart of distribution of survey result.

2. An important question is that some issues is indirected, such as Promoting road safety understanding in Thailand (Knowledge) seem unrelated to satisfaction. Is it really reasonable to directly compare horizontally based on satisfaction? 

Comments on the Quality of English Language

 Overall, English is fluent, the intial manuscript only requires slight revision.

Author Response

I appreciate your dedicated effort in reviewing this manuscript.  Please find the detailed responses below and the corresponding revisions/corrections highlighted in the re-submitted files. 

Reviewer 2 Report

Comments and Suggestions for Authors

The article explores the use of stickers for social awareness and road safety promotion. Different stickers were designed for this purpose and disseminated through the Line application. This is an interesting and novel topic, representing an unusual form of communication campaign but which could have relevant practical applications. I will now make some recommendations that I consider could improve the quality of the paper.

In relation to the introduction and the literature review, the problem is presented to the reader, however I consider that it should be further developed with respect to some of the ideas mentioned. For example, I recommend including a subsection that synthesizes the existing evidence on the use of social networks for the promotion of preventive campaigns in the traffic or public health sector (e.g. https://doi.org/10.3389/fpubh.2022.1045645, https://doi.org/10.1145/2479724.2479743, etc). I also recommend including the objectives of the study and the main hypotheses of the research in a specific subsection.

Regarding the methodology and results, they are clear and adequately presented. But, I suggest further developing the discussion section. This section, although integrated with the conclusions, should not only be a summary of the results but should contrast the data obtained with other research. Thus, I recommend that this section be developed by answering the questions: were the results in accordance with expectations, are the results congruent with other similar research? And, if not, what elements explain the discrepancies that have occurred? A specific subsection should also be included in which the limitations of the study and future lines of research are presented.

I congratulate the authors on the novelty of their research and recommend publication of the manuscript, once the suggested minor modifications have been made.

Author Response

(The authors gave the same response as above.)

Reviewer 3 Report

Comments and Suggestions for Authors

The authors presented research that aimed to create Line application stickers, a top chat platform for Thai, using the Participatory Design (PD) approach. A very interesting approach for the preparation of preventive and educational materials (Line Stickers) to promote Road Safety. Although the case study is from Thailand, it can easily be applied to other countries.

This manuscript is concise and clear in purpose and organization. Research methodology is based on the PD approach, involving users as stakeholders directly in the design process. The authors described in detail the results of the conducted research and gave their conclusions.

Although I think this is good research, I suggest the authors add in the paper a little more detail about the implementation of Line Stickers: Are these Line Stickers already used in practice? If they are used, how long have they been used? If they are not used, when is the implementation planned? How and where are the Line Stickers distributed (e.g., schools, driving schools)? Where are they available? Do the authors have information about the acceptance of Line Stickers by users? Which ones, younger, or older?

Author Response

(The authors gave the same response as above.)

Reviewer 4 Report

Comments and Suggestions for Authors

This paper aims to design and develop Line stickers for a road safety campaign in Thailand, and provides insights into the factors influencing the effectiveness of the stickers. In general, this study displays remarkable novelty and significant practical relevance. The comments are as follows:

i) Despite the meaningful and visionary aspects of this work, the abstract and introduction fail to effectively highlight your key contributions. It is recommended that the paper expounds upon the relevance of this study to traffic safety, specifically the impact of emojis on people's attitudes towards traffic safety and their potential positive impact on children.

ii) Figure 1 provides insufficient information to the readers, and it may be advisable to either remove or optimize it.

iii) The survey and interview data are suggested to visualize them using figures to visually present key information to the readers in a more intuitive manner.

iv) Is the selection of participants representative? It needs to be further elaborated upon with careful consideration.

v) If possible, providing some draft versions of stickers would further highlight the effort put into this work.

vi) The reliability and validity of the satisfaction survey need to be introduced.

Author Response

(The authors gave the same response as above.)

Round 2

Reviewer 1 Report

Comments and Suggestions for Authors

The authors have well solved my comments. It can be published with this version.

Comments on the Quality of English Language

English is fluent, no additional revision is required.

Reviewer 2 Report

Comments and Suggestions for Authors

The authors have taken into account the suggestions I provided in my previous review, so I consider that the manuscript is suitable for publication.

Reviewer 3 Report

Comments and Suggestions for Authors

In this new version, the authors answered all my previous comments and added the information needed. I have no new comment to add. 

Reviewer 4 Report

Comments and Suggestions for Authors

All the reviewer's comments have been carefully addressed, and the paper is recommended for publication.